# Association between Sexual Activity during Pregnancy, Pre- and Early-Term Birth, and Vaginal Cytokine Inflammation: A Prospective Study of Black Women

**DOI:** 10.3390/healthcare11141995

**Published:** 2023-07-11

**Authors:** Kylie Dougherty, Yihong Zhao, Anne L. Dunlop, Elizabeth Corwin

**Affiliations:** 1School of Nursing, Columbia University, New York, NY 10032, USAyz2135@cumc.columbia.edu (Y.Z.); 2School of Medicine, Emory University, Atlanta, GA 30322, USA; amlang@emory.edu

**Keywords:** vaginal sex, preterm birth, early term birth, cytokines, inflammation

## Abstract

This study aimed to investigate the association between sexual activity during pregnancy and adverse birth outcomes among Black women, and to explore whether vaginal cytokine inflammation mediates this association. Data from 397 Black pregnant women through questionnaires on sexual activity and vaginal biosamples during early (8–14 weeks) and late (24–30 weeks) pregnancy, and birth outcomes were analyzed. Using a data-driven approach, the study found that vaginal sex during late pregnancy was associated with spontaneous early-term birth (sETB, 38–39 completed weeks’ gestation) (OR = 0.39, 95% CI: [0.21, 0.72], *p*-value = 0.003) but not with spontaneous preterm birth (sPTB) (OR = 1.08, *p*-value = 0.86) compared to full-term birth. Overall, despite vaginal sex in late pregnancy showing an overall positive effect on sETB (total effect = −0.1580, *p*-value = 0.015), we observed a negative effect of vaginal sex on sETB (indirect effect = 0.0313, *p*-value = 0.026) due to the fact that having vaginal sex could lead to elevated IL6 levels, which in turn increased the odds of sETB. In conclusion, the study found an overall positive association between sexual activity on ETB and a negative partial mediation effect via increased vaginal cytokine inflammation induced by vaginal sexual activity. This inconsistent mediation model suggested that vaginal sexual activity is a complex behavior that could have both positive and negative effects on the birth outcome.

## 1. Introduction

Preterm birth (PTB) and early-term birth (ETB) are significant public health concerns in the United States, particularly among Black women, and they result in high costs to the healthcare system [1,2,3]. PTB is defined as a birth occurring prior to 37 completed weeks’ gestation [1], while ETB is defined as a birth occurring between 37 weeks and 38 weeks, 6 days. Infants born before full-term may not be fully developed, as they miss out on significant growth and development during the final weeks of gestation [4]. Children born preterm have an increased risk of developmental and chronic conditions such as motor and cognitive neurodevelopmental delays and disabilities, chronic lung disease, and respiratory problems [1,5,6,7,8,9,10,11,12,13]. They are also less likely to perform at grade level in elementary school [4,11,14,15]. Infants born early-term have a higher risk of death, lower birth weight, and are more likely to spend time in intensive care units [4,16]. Therefore, it is crucial to identify the risk and protective factors of PTB and ETB to improve maternal and infant health.

Among the known risk factors for preterm birth (PTB) in the United States, one of the most significant is maternal race, particularly the Black race [17]. According to the most recent US Vital Statistics report on PTB, the overall PTB rate in the country is 10.1% [18]. However, when we consider racial breakdown, a more nuanced picture emerges, revealing a PTB rate of 14.4% among Black women, which is over 50% higher than the rate of 9.1% among White women. This disparity is most pronounced for spontaneous preterm birth (sPTB), occurring four times more frequently in Black women (8.9%) compared to White women (2.2%). Consequently, this contributes to the higher incidence of low birth weight among Black infants, who are twice as likely as White infants to have low birth weight (14% vs. 7%) and nearly 2½ times more likely to experience infant mortality [19]. While low socioeconomic status (SES) is recognized as a risk factor for PTB, studies consistently demonstrate that SES alone explains less than half of the Black–White disparity in PTB [1,20]. 

Although many factors contribute to PTB and ETB, the impact of vaginal sex during pregnancy on these outcomes remains unclear [21]. Previous research has yielded mixed findings, with some studies suggesting that vaginal sex during late pregnancy [22] increases the risk of premature labor [23,24], while others have found no effect [25,26] or even a reduced risk of premature delivery [27,28]. Most prior research has focused on the relationship between vaginal sex and PTB, with little known about the connection between vaginal sex and ETB. Given the adverse health outcomes associated with ETB, it is essential to investigate the vaginal sex–ETB relationship. At the biological level, previous studies have established associations between vaginal and systemic proinflammatory cytokines, such as interleukin-1beta (IL-1β), tumor necrosis factor alpha (TNFα), interferon-gamma (IFN-γ), and interleukin-6 (IL-6), and the incidence of PTB [29,30,31,32,33,34]. Cytokines are proteins secreted by white blood cells that interact with other cells, typically triggering either a pro-inflammatory (innate) or anti-inflammatory (adaptive immune) response within the body [35,36]. These cytokines are present both systemically and locally within the vaginal canal [23]. It is postulated that increased inflammation within the vagina, and potentially systemically, promotes cervical ripening and uterine contractility while increasing decidual and membrane activation [31,33]. The triggers that lead to heightened inflammation, however, are not fully understood [37]. Our hypothesis is that vaginal sex during late pregnancy is likely to increase vaginal inflammation, resulting in an increased risk of premature delivery. This study has the potential to identify potential mediating roles of novel biobehavioral triggers in the relationships between vaginal sex and premature delivery.

The current study aims to investigate whether engaging in sexual behavior during pregnancy increases the risk of spontaneous preterm birth (sPTB) and spontaneous early-term birth (sETB) among a socioeconomically diverse sample of pregnant Black women. Additionally, we examine whether vaginal pro- and anti-inflammatory cytokine concentrations may mediate the relationship between sexual activity during late pregnancy and adverse birth outcomes. 

## 2. Methods

*Overview.* This study is a secondary analysis of a 5-year prospective longitudinal study conducted in Atlanta, Georgia. The original study enrolled over 500 healthy, Black women in their first trimester of pregnancy and followed them through delivery, as described in detail elsewhere [38]. Data were collected prospectively during routine prenatal care visits at two time points: the first visit occurred between 8 and 14 weeks’ gestation and the second visit between 24 and 30 weeks’ gestation. This study design allowed for the assessment of various factors, including vaginal sex and inflammation markers, in real-time as they occurred during the participants’ pregnancies. Medical records were used to determine infant gestational age at birth using the American College of Obstetricians and Gynecologists (ACOG) algorithm, which combines ultrasound and last menstrual period information [39]. In this secondary data analysis project, we included only these who had spontaneous early-term birth, spontaneous preterm birth, or full-term birth, resulting in 397 participants in the final sample.

*Ethics Approval.* This is a secondary data analysis study, and the institutional review board ethics approval was obtained for the original study [38].

*Selection Criteria.* Participants were included in the original study [38] if they (1) self-identified as African American, (2) were between 8 and 14 weeks’ gestation (gestational age was confirmed by clinical records and/or ultrasound), (3) were able to comprehend written and spoken English, (4) were between 18 and 40 years of age, and (5) were not experiencing any chronic medical conditions or taking prescriptions for chronic conditions (verified by prenatal records). Additionally, participants needed to live within a 20-mile radius of the laboratory to decrease the amount of time spent transporting the biological samples, thus preserving the validity of the biomarkers.

*Biological Samples and Laboratory Assays.* The study analyzed biological samples collected during regularly scheduled prenatal care visits for subsequent measurement of cytokine levels using the MesoScale assay platform [40], and C-reactive protein (CRP) using the enzyme-linked immunosorbent assay (ELISA) [41]. The collection, processing, and assay details have been previously published [38]. Briefly, after providing informed consent, the study coordinator escorted each woman individually to a private room, where they were provided both verbal and pictorial instructions on self-collecting vaginal swabs. After answering any questions, the research coordinator stepped out of the room while the woman donned sterile gloves and then privately inserted the swab approximately 3–4 inches from the introitus, into the midportion of the vaginal vault. Vaginal sampling was conducted using a sterile Catch-AllTM Sample Collection Swab (Epicentre Biotechnologies, Madison, WI, USA). After swabbing, and without touching the top of the swab, the woman immediately handed it to the research coordinator who then placed the swab in a MoBio bead tube (moGio Laboratories, Inc. San Diego, CA, USA). This tube was clearly labeled and kept on ice until transferal to the lab where it was immediately stored at −80° until assayed, including for cytokines using standard enzyme-linked immunoassay (ELISA) kits [41]. The concentration of C-reactive protein (CRP) was additionally determined via the four parameter calibration curves identified using the BioTek Gen5 software version 2.0.9.

The cytokine panel was selected to include both pro- and anti-inflammatory cytokines as well as IL-6, a myokine that promotes both a pro- and anti-inflammatory activity [36,42,43,44]. INFγ is a key anti-inflammatory cytokine critical to innate and adaptive immunity [45]. TNF-α is an inflammatory cytokine produced during acute inflammation [46]. Interleukin-10 (IL-10), in contrast, inhibits the host inflammatory immune response, thereby mitigating tissue damage and immunopathology. 

*Questionnaire Data.* At each study visit, participants completed surveys about whether they had engaged in vaginal sex within the past month. The data were dichotomized to indicate whether the participant reported having vaginal sex. 

*Clinical Data.* The research team used a standardized chart abstraction tool to extract clinical data from medical records, including gestational age at birth. Gestational age at birth (in completed gestational weeks) was determined from the delivery record using the best obstetrical estimate [47], which was based on early pregnancy dating by last menstrual period (LMP) and/or early ultrasound, as required for study enrollment. All participants received early pregnancy dating by last menstrual period (LMP) and/or early ultrasound, given enrollment criteria. The type of labor (spontaneous, induced, none), mode of delivery (vaginal, cesarean section), and indication for induction and/or cesarean section were also recorded and used to phenotype birth outcomes. This study focused on three groups of birth outcomes: spontaneous preterm birth (between 22-0/7 and 36-6/7 weeks gestation), spontaneous early-term birth (between 37-0/7 and 38-6/7 weeks gestation), and full-term birth (39-0/7 weeks or greater), which served as the control group.

*Statistical Analysis.* Descriptive statistics were used to summarize sample characteristics, including frequencies (for categorical variables), and mean and standard deviation (for continuous measures) using R software version 4.1.1. Bivariate associations between demographic variables and birth outcomes (sPTB vs. full-term birth (FTB) and sETB vs. FTB) were evaluated using two-sample T-tests (for continuous measures such as age) and chi-square tests (for categorical measures such as education level). In this study, we focused on ratios between pro-inflammatory (i.e., INFγ, TNF, CRP) cytokines and IL-6 to anti-inflammatory (i.e., IL-10) cytokines, as a balanced ratio of pro- and anti-inflammatory cytokines is essential to regulate the maternal inflammation system throughout pregnancy [48]. Although the log-transformed ratios were used in the analysis, results from models with the inflammatory level at the original scale remained similar. Multiple hypothesis tests were performed in assessing bivariate associations. To adjust for potential inflated false discoveries due to multiple comparison problems, we controlled the False Discovery Rate [49] to the 0.05 level in this study. 

Logistic regression was used to examine the relationship between vaginal sex and birth outcomes. Residual plots and the Shapiro–Wilk test were used to check for potential violation of the model assumptions. Subsequent mediation analysis focused only on the effect of vaginal sex during late pregnancy on sETB as this was the only significant association found. A two-step analytical approach [50] was used to assess whether vaginal inflammation mediates the effect of vaginal sex on sETB. 

In Step 1, the iterative Random Forest (iRF) algorithm [51] was used to identify the most important pro- and anti-inflammatory cytokine ratios for predicting sETB. The iRF algorithm computed a variable importance score for each cytokine measure, with a larger mean decrease in the Gini impurity index (GI) indicating greater importance in predicting sETB. In this study, the vaginal IL6/IL10 ratio was identified as the top predictor of sETB. In Step 2, potential mediating roles of IL-6/IL-10 cytokine levels during late pregnancy on the effects of vaginal sex were examined using a causal mediation analysis approach. Causal mediation analyses [52] were performed using functions in R package meditation [53]. We controlled potential confounding effects due to age, sex of newborn, parity, marital and cohabitating status, and education level in the analyses. 

## 3. Results

### 3.1. Sample Characteristics

Of the 397 participants with exposure and outcome data for inclusion in this study, there were 49 cases of sPTB, 93 cases of sETB, and 255 FTB. Table 1 presents the sociodemographic and clinical characteristics of the study sample. There were no statistically significant inter-group differences (sPTB vs. FTB, and sETB vs. FTB) in mother’s age, educational level (high school or less versus other), body mass index (BMI) at the first prenatal visit, and marital or cohabitating status. However, women in the sETB group had a higher proportion of prior birth experience (n = 62, 66.7%) than those in the FTB group (n = 123, 48.2%; adjusted *p*-value = 0.012). In addition, fewer women in the sPTB group carried a female fetus (n = 14, 28.6%) than those in the FTB group (n = 135, 52.9%; adjusted *p*-value = 0.012).

### 3.2. Linking Sexual Behavior and Inflammation with Birth Outcomes 

At early pregnancy, 77.6% (n = 38) of sPTB women, 66.7% (n = 62) of sETB women, and 74.9% (n = 191) of FTB women self-reported having vaginal sex. The proportion of women having vaginal sex during late pregnancy reduced to 51% (n = 25), 44.1% (n = 41), and 60% (n = 153) among sPTB, sETB, and FTB women, respectively. Controlling for the confounding factors listed in the Statistical Analysis section, subsequent logistic regression analyses revealed that having vaginal sex during late pregnancy was associated with a 61% decrease in the odds of having sETB (OR = 0.39, 95% CI: [0.21, 0.72], adjusted *p*-value = 0.01). sETB was marginally associated with vaginal sex at the early stage after multiple-comparison adjustment (OR = 0.51, 95% CI: [0.29, 0.93], *p*-value = 0.02, adjusted *p*-value = 0.07). In contrast, sPTB was not associated with vaginal sex at either early (OR = 0.91, 95% CI: [0.41, 2.15], adjusted *p*-value = 1.0) or late (OR = 1.08, 95% CI: [0.44, 2.87], adjusted *p*-value = 1.0) pregnancy. 

Descriptive statistics (mean +/− sd) for cytokine measures by gestational age at birth outcomes can be found in Appendix A. Iterative Random Forest analysis results (see Figure 1) showed that the vaginal log ratio of IL-6/IL-10 at late pregnancy was the most important inflammatory predictors of sETB. Therefore, IL6/IL10 was regarded as the potential mediator in the subsequent analyses. 

### 3.3. Mediating Role of the Vaginal IL-6/IL-10 Ratio

To explore the potential mediating role of the vaginal IL-6/IL-10 ratio on the relationship between vaginal sexual behavior and sETB, we performed mediation analysis (Figure 2). Our analysis indicated that both vaginal sexual behavior and the ratio of IL-6/IL-10 during late pregnancy were significantly associated with sETB status. Briefly, mediation analysis decomposed the effect of vaginal sexual behavior on sETB into indirect and direct effects, with the indirect effect explaining the effect of sexual behavior on sETB working through the IL-6/IL-10 ratio and the direct effect explaining the effect of sexual behavior on sETB working through other mechanisms.

Figure 2 shows the mediation analysis results, and the regression coefficients from these models can be found in Table 2. First, there was a significant overall positive effect of vaginal sex during late pregnancy on sETB (β = −0.944, OR = 0.39, *p*-value = 0.003). Controlling for potential confounding effects due to age, sex of newborn, parity, marital and cohabitating status, and education level, our model indicated that the odds of having sETB for women with vaginal sex at late pregnancy were 61% lower than those for women without vaginal sex. Second, in the mediation pathway, we observed that having vaginal sex was associated with a 0.63-unit increase in log ratio of IL-6/IL-10 (β = 0.63, *p*-value = 0.036). Furthermore, a one-unit increase in the log ratio of IL-6/IL-10 was associated with an increase in the odds of having sETB by 19.9% (β = 0.1812, *p* = 0.02). Mediation analysis suggested that engaging in vaginal sex could lead to an increase in IL-6 levels, which, in turn, could contribute to an increased likelihood of experiencing sETB.

In summary, although engaging in vaginal sex was associated with an overall significant reduction in the odds of experiencing sETB (total effect = −0.1560, *p*-value = 0.015), it is important to recognize the potential offsetting effects (indirect effect = 0.0313, *p*-value = 0.026, proportion mediated = 18.90%). This implies that approximately 18.9% of the total effect of vaginal sex on sETB can be attributed to the mediating role of IL-6, indicating a partial mediation effect in the relationship between vaginal sex and sETB. It is understandable that IL-6, as an inflammatory biomarker, may not completely explain the relationship between vaginal sex and sETB, given that sETB has various causes. Therefore, it is crucial to recognize the multifactorial nature of sETB and consider other contributing factors in future investigations.

## 4. Discussion

The study investigated the complex relationship between vaginal sex during early and late pregnancy and obstetric outcomes, especially sPTB and sETB in healthy Black women with singleton pregnancies. Our findings contribute to the understanding of this intricate association, with a particular focus on the potential mediating role of vaginal inflammation. 

First, the relationship between vaginal sex during pregnancy and sPTB has been investigated in previous studies, and our findings of no significant relationship between vaginal sex and sPTB are consistent with some of these studies [25,26]. However, other studies have reported mixed findings on the relationship between vaginal sex and sPTB [24,54]. Given that there were only 49 cases of sPTB in our sample, the study might not have sufficient power to detect a significant vaginal sex and sPTB association with a small to moderate effect size. Therefore, caution should be exercised in interpreting our results. 

Second, this analysis builds upon the literature surrounding the influences on sETB, which is a phenomenon that impacts the health of newborns but is not studied as often as sPTB. The full impact of vaginal sex on ETB requires further research. Toward this goal, our study tested two hypotheses: (1) whether vaginal sexual behavior during pregnancy increases the odds of having sETB, and (2) whether the negative impact of vaginal sexual behavior on sETB is mediated by the inflammatory biomarkers. Our analysis showed an overall significant positive effect of vaginal sex during late pregnancy on sETB, indicating a reduced risk of having sETB. Interestingly, mediation analysis suggested that vaginal sexual behavior could have a negative impact on sETB, as vaginal sex during late pregnancy was associated with an increased vaginal IL-6/IL-10 ratio, while increased vaginal IL-6/IL-10 ratio was linked to an increased risk of sETB. 

The mediation analysis result was in line with our original hypothesis that vaginal sex has a negative impact on the sETB. Previous research has established the importance of IL-6 in triggering spontaneous preterm birth, indicating that the elevation of IL-6 level may be involved in regulating the onset of labor [55] through its positive correlation with the expression of genes controlling prostaglandin synthesis and signaling [56]. Recent studies have also reported elevated cervico-vaginal fluid concentrations of IL-6 among women with preterm delivery [57]. Our study adds to this body of evidence by linking vaginal sex during late pregnancy with an increase in vaginal IL-6/IL-10 ratio, which in turn results in an increased risk of having sETB. Taken together, these findings shed light on the significance of IL-6 as a potential biological pathway through which vaginal sex influences the likelihood of experiencing sETB. 

It is important to note that the overall significant positive (i.e., beneficial) effect of vaginal sex on sETB was contradictory to what we have initially hypothesized. However, this seemingly contradictory result was not surprising and explainable. In this study, we have an inconsistent mediation model [58] (i.e., positive indirect effect and negative direct effect, see Figure 2). In the field of medical research, it is not uncommon to encounter inconsistent mediator effects, which can be particularly significant when assessing counterproductive effects of experiments. In such cases, the manipulation involved may have resulted in contradictory mediated effects. For a more comprehensive understanding of this topic, interested readers are encouraged to refer to the work of Mackinnon et al. [58] and the references provided therein for valuable insights into the complexities associated with inconsistent mediator effects in medical research. Briefly, relevant to our study, inconsistent mediator effects can occur when there are different positive and negative pathways through which vaginal sexual activity influences sETB. Indeed, the relationship between sexual intercourse and obstetric outcomes is multifaceted, likely influenced by various factors beyond vaginal inflammation. While our study focused only on the role of vaginal inflammation, it is crucial to acknowledge the potential contribution of other factors in the complex relationship between vaginal sexual behavior and birth outcomes. For example, previous research has suggested that factors such as short cervical length [59,60], poor nutritional status [61], higher parity [62,63], and lower socioeconomic status [64] could impact birth outcomes. Health benefits of vaginal sexual activity on a wide range of medical and psychological measurements have also been reported [65]. As demonstrated by our inconsistent mediation model, this investigation suggested that there might be multiple potential mechanisms by which vaginal sexual activity positively or negatively influences the birth outcome. Therefore, to provide a more comprehensive understanding of the relationship between sexual intercourse and obstetric outcomes, future studies should take into consideration a broader set of risk or protector confounders/mediators, enabling a more nuanced interpretation of the association between vaginal sex and birth outcomes. 

Third, in terms of the significance of our study, it is important to emphasize its contribution to the existing literature. While previous research has investigated the relationship between vaginal sex and preterm birth, our study specifically focused on healthy Black women with singleton pregnancies, a population that experiences significant health disparities in reproductive health. By addressing this critical gap in the literature, our study provides valuable insights into the unique context of sexual activity during pregnancy in this population. Furthermore, our study incorporated longitudinal biological samples and utilized a two-step big data approach [50], highlighting the potential utility of this methodology for identifying novel findings in obstetrics and gynecology research. This approach has been successfully applied in various fields [66,67,68] to accelerate scientific discovery. These strengths enhance the scientific rigor of our findings and contribute to advancing knowledge in the field.

Finally, one major limitation of our study is that the main exposure variable, vaginal sexual behaviors, was coded as a binary variable. It has been reported that vaginal cytokine levels can be affected by various sexual practices such as the number of partners, frequency of vaginal sex, and unprotected vaginal intercourse [69,70,71]. Although we collected data on the frequency of vaginal sexual behaviors at each trimester, there were missing data on these measures due to participants ‘skipping’ or not providing an answer to this item, precluding our ability to study the dose–response relationship between vaginal sexual behavior and the risk of sETB. Future research should explore the role of vaginal sex on ETB with vaginal sex as a frequency as opposed to a binary variable. Exploring vaginal sex through the lens of frequency will provide a more granular understanding of the influence of vaginal sex in terms of whether there are any potential differences in frequent or occasional vaginal sex during the later weeks of pregnancy on gestational age at birth outcomes. Additionally, factors such as the number of partners and unprotected versus protected vaginal sex could also influence the role of vaginal sex overall on ETB and should be explored in future research. Finally, the study sample was recruited from pregnant US-born women who self-identified as African American and who were receiving care from two hospitals in Atlanta, GA. Therefore, the study sample is not representative of the African American population in the United States. The generalizability of our findings should be tested in other independent studies. 

## 5. Conclusions

In conclusion, our study reveals the complex relationship between vaginal sex during pregnancy and obstetric outcomes, with a particular emphasis on the potential mediation effect of vaginal inflammation. Our findings highlight the need to consider various factors beyond vaginal inflammation, including short cervical length, nutritional status, parity, and socioeconomic status, when investigating this relationship. The partial mediation effect observed suggests that the impact of vaginal sex on early-term birth is influenced by multiple factors, emphasizing the need for further research. While our findings contribute to the understanding of this complex association, the message to take home is that the safety of vaginal sex during pregnancy is still an area that requires further investigation. Future studies should aim to explore additional factors, employ more comprehensive measures of sexual activity, and involve diverse populations to provide a more nuanced understanding of the relationship between sexual intercourse and obstetric outcomes.

Our study makes a significant contribution to the existing literature by emphasizing the multifaceted nature of the relationship between sexual activity during pregnancy and birth outcomes. While previous studies have explored this association, our research extends beyond a binary understanding by elucidating the dual impacts on birth outcomes. The findings of both positive and negative contributions of sexual activity on birth outcomes provide valuable insights into the complexity of this topic.

Of particular significance is our identification of a potential pathway through which sexual activity may negatively impact birth outcomes, namely, the association with vaginal inflammation. Although the precise biological mechanisms underlying the positive contributions of sexual activity on birth outcome remain unclear, our study highlights the importance of considering inflammation as a potential mediator in future investigations.

Overall, our findings underscore the need for comprehensive and individualized discussions regarding sexual activity during pregnancy. Health practitioners and expectant individuals should be aware of the nuanced nature of this topic and consider factors such as personal medical history, gestational age, and the presence of vaginal inflammation. Further research is warranted to uncover the underlying biological mechanisms and to develop tailored guidelines that can better inform clinical practice.

## Figures and Tables

**Figure 1 healthcare-11-01995-f001:**
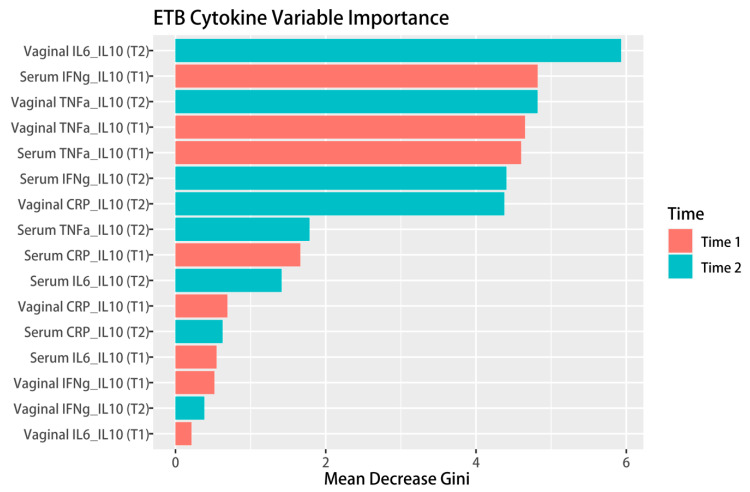
Random forest analyses showed that IL6/IL10 ratio during late pregnancy was the most important cytokine predictor of sETB. Time 1 and Time 2 stand for early and late pregnancy, respectively.

**Figure 2 healthcare-11-01995-f002:**
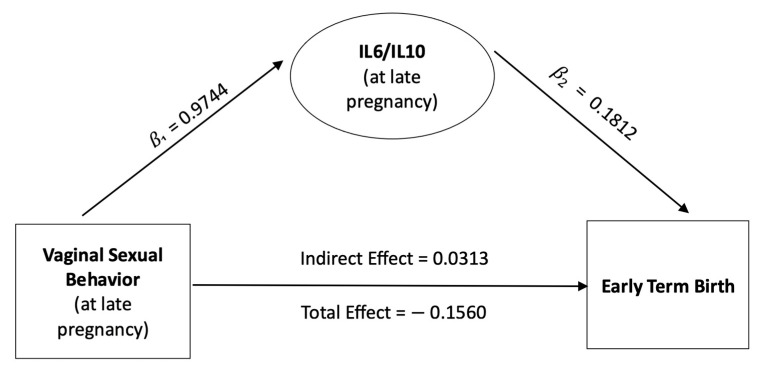
Mediation effect of vaginal IL6/IL10 ratio on relationship between vaginal sexual behavior and sETB. $\beta_{1}$ = regression coefficient to assess relationship between vaginal sexual behavior and IL6/IL10 ratio during late pregnancy. $\beta_{2}$ = regression coefficient to assess relationship between IL6/IL10 and sETB, controlling for vaginal sexual behavior. Indirect Effect = the effect of vaginal sexual behavior on sETB that works through elevated IL6/IL10 ratio.

**Table 1 healthcare-11-01995-t001:** Demographic characteristics by gestational age at birth outcome.

Variables	P sPTB	sETB	FTB
Sample Size	49	93	255 (referent)
Age (years)	24.51 ± 4.796	25.31 ± 4.83	24.75 ± 4.65
Education (High School or less)	32 (65.3%)	55 (59.1%)	125 (49%)
BMI at 1st prenatal visit	26.24 ± 6.29	28.08 ± 7.86	28.93 ± 7.68
Married or Cohabitating	32 (65.3%)	46 (49.5%)	134 (52.5%)
Prior Birth ^1^	30 (61.2%)	62 (66.7%)	123 (48.2%)
Baby’s sex (Female) ^2^	14 (28.6%)	52 (55.9%)	135 (52.9%)

^1^: Women in the sETB group had a significantly higher proportion of prior birth experience than those in the FTB group (adjusted *p*-value = 0.012). ^2^: Women in the sPTB group had a significantly smaller proportion of carrying a female baby than those in the FTB group (adjusted *p*-value = 0.012).

**Table 2 healthcare-11-01995-t002:** Coefficients from logistics regression models with sETB as the outcome.

	Model 1				Model 2			
Variable	Estimate	SE	z.Value	*p*-Value	Estimate	SE	z.Value	*p*-Value
(Intercept)	−0.91	1.12	−0.81	0.416	−1.38	1.27	−1.08	0.278
age	0.03	0.03	0.98	0.329	0.04	0.04	1.05	0.296
Baby Sex (Female)	0.22	0.29	0.75	0.450	0.19	0.32	0.59	0.553
Prenatal BMI *	−0.04	0.02	−1.78	0.074	−0.03	0.02	−1.48	0.138
Parity	0.76	0.32	2.36	0.018	0.67	0.36	1.87	0.061
≤High School	0.61	0.32	1.92	0.054	0.43	0.35	1.23	0.218
Married and Cohab (Yes)	−0.66	0.31	−2.15	0.031	−0.63	0.33	−1.88	0.060
Vaginal Sex	−0.94	0.31	−3.00	0.003	−1.03	0.35	−2.92	0.003
IL6_IL10					0.18	0.08	2.32	0.020

* BMI at the first prenatal visit.

## Data Availability

The data presented in this study are available on request from the corresponding author. The data are not publicly available due to restrictions on written permission for the use of the data.

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
