# Peer review of "Association between Sexual Activity during Pregnancy, Pre- and Early-Term Birth, and Vaginal Cytokine Inflammation: A Prospective Study of Black Women"

_healthcare, 2023, doi:10.3390/healthcare11141995_

Round 1

Reviewer 1 Report

Thank you for inviting me to review this work. My specific comments regarding this manuscript are below:

- All abbreviations should have been provided in full on the first mention and this applies to the title, running (short) title, abstract, impact statement, main text, and each table/figure independently as they will be read independently. Please use the abbreviations correctly and effectively. So, all the abbreviations should be checked (i.e. FTB, BMI, iRF in the results section, etc.).

- A manuscript title is a phrase that serves two important functions: to identify the main topic or point of the paper and to attract readers. The words in the title may determine whether readers can find your article in a database with search terms. I consider that the title did not truly reflect the hypothesis of this manuscript because of the absence of words regarding preterm birth. Please select a title that accurately describes the contents of your manuscript.

- Some of the references are not current. Please cite recently published studies.

- How did you design your study (prospectively, retrospectively, etc.)? Please write it in the methods section.

- The inclusion and exclusion criteria of the participants are not clear. It should be specified. Please write the features of the study group more comprehensively.

- The authors reported that vaginal sex during late pregnancy (occurring within one month during the period of 24-30 weeks’ gestation) was associated with a reduced risk of sETB, indicating an overall beneficial effect of vaginal sex on early term birth. However, mediation analysis revealed that vaginal sex during late pregnancy was also associated with an increased vaginal IL6/IL10 ratio, which in turn was linked to an increased risk of sETB. How could it be possible that this result suggests that vaginal sex during late pregnancy could have a partial mediation effect on sETB, with both positive and negative impacts on early-term birth? According to the analysis, vaginal sex during late pregnancy was associated with a reduced risk of sETB.

- Also, the discussion section of the manuscript is insufficient. Please discuss your findings; explain the significance of those results and tie everything back to the research question. Interpretation of the results is inconsistent.

- Practical implications and future research direction are not mentioned. Please discuss the generalisability (external validity) of the study results.

- In conclusion, clarify the message that the reader will take home. Is vaginal sex safe or not? Moreover, lots of articles in the literature investigated the relationship between vaginal sex and preterm birth. What is your contribution to the current literature?

- There are several typographical errors that should need to be corrected (i.e. factor-alpha, we hypothesize that vaginal sex, this stıdy has the potential, early-term birth, etc.).

Author Response

We highly appreciate the reviewers’ constructive and valuable comments which significantly improve our manuscript. We now provide point-to-point response to the comments, with the changes highlighted in RED in the manuscript.

Reviewer 1 Comments and Suggestions for Authors

Thank you for inviting me to review this work. My specific comments regarding this manuscript are below:

  1. All abbreviations should have been provided in full on the first mention and this applies to the title, running (short) title, abstract, impact statement, main text, and each table/figure independently as they will be read independently. Please use the abbreviations correctly and effectively. So, all the abbreviations should be checked (i.e. FTB, BMI, iRF in the results section, etc.).

Response: Thank you for bringing this to our attention. We have carefully checked all abbreviations and made the changes accordingly. All changes are marked in red.

  1. A manuscript title is a phrase that serves two important functions: to identify the main topic or point of the paper and to attract readers. The words in the title may determine whether readers can find your article in a database with search terms. I consider that the title did not truly reflect the hypothesis of this manuscript because of the absence of words regarding preterm birth. Please select a title that accurately describes the contents of your manuscript.

Response: Thank you so much for the suggestion. We have changed the title to “Association between Sexual Activity during Pregnancy, Pre- and Early-term Birth, and Vaginal Cytokine Inflammation: A Prospective Study of Black Women”.

  1. Some of the references are not current. Please cite recently published studies.

Response: Thank you for the suggestion. We have included several new references in the revised version.

  1. How did you design your study (prospectively, retrospectively, etc.)? Please write it in the methods section.

Response: Thank you for the question. Our study should be considered as a prospective study for the following reasons:

Our study utilized an existing dataset where biosamples were collected across two time points, and participants were surveyed about their sexual activity in the past 30 days at each time point. Prospective studies involve collecting data on exposure variables and outcomes in real-time, as they occur over a defined period. In our study, the assessment of vaginal sex and biosamples collection occurred contemporaneously with the participants' current experiences, making it a prospective study.

While our study analyzed retrospective information on sexual activity in the past 30 days, it is important to note that the biosample collection and birth outcome assessment were conducted prospectively. We used the available data to examine the relationship between vaginal sex, inflammation markers measured in biosamples, and the subsequent birth outcome. This approach allows us to investigate temporal associations and explore potential causal pathways between these variables.

Therefore, while retrospective information on sexual activity was collected, our study can be appropriately categorized as a prospective study due to the timing of biosample collection and birth outcome assessment, which aligns with the core principles of prospective study design.

We appreciate the reviewer's inquiry and the opportunity to clarify this aspect of our study design. We have revised the Overview paragraph in the Methods section accordingly (see below).

“Overview. This study is a secondary analysis of a 5-year prospective longitudinal study conducted in Atlanta, Georgia. The original study was conducted between 2014 and 2018 and enrolled over 500 healthy, Black women in their first trimester of pregnancy and followed them through delivery, as described in detail elsewhere.33 Data were collected prospectively during routine prenatal care visits at two time points: the first visit occurred between 8-14 weeks gestation, and the second visit between 24-30 weeks’ gestation. This study design allowed for the assessment of various factors, including vaginal sex and inflammatory markers, in real-time as they occurred during the participants' pregnancies. Medical records were used to determine infant gestational age at birth using the American College of Obstetricians and Gynecologists (ACOG??) algorithm, which combines ultrasound and last menstrual period information.34” 

  1. The inclusion and exclusion criteria of the participants are not clear. It should be specified. Please write the features of the study group more comprehensively.

Response: Thanks for the comments. We had added the following inclusion and exclusion criteria to the Methods section.

“Selection Criteria. Participants were included in the original study33 if they 1) self-identified as African American, 2) were between 8-14 weeks gestation (gestational age was confirmed by clinical records and/or ultrasound), 3) able to comprehend written and spoken English, 4) between 18-40 years of age, and 5) were not experiencing any chronic medical conditions or taking prescriptions for chronic conditions (verified by prenatal records). Additionally, participants needed to live within a 20-mile radius of the laboratory, to decrease the amount of time spend transporting the biological samples, thus preserving the validity of the biomarkers.”

6.The authors reported that vaginal sex during late pregnancy (occurring within one month during the period of 24-30 weeks’ gestation) was associated with a reduced risk of sETB, indicating an overall beneficial effect of vaginal sex on early term birth. However, mediation analysis revealed that vaginal sex during late pregnancy was also associated with an increased vaginal IL6/IL10 ratio, which in turn was linked to an increased risk of sETB. How could it be possible that this result suggests that vaginal sex during late pregnancy could have a partial mediation effect on sETB, with both positive and negative impacts on early-term birth? According to the analysis, vaginal sex during late pregnancy was associated with a reduced risk of sETB.

Response: Thank you for this important question. Our findings showed that while vaginal sex during late pregnancy is associated with a reduced risk of sETB overall, there is an additional pathway through which it could have a partial mediation effect, leading to an increased risk of sETB. This complexity highlights the intricate interplay between various biological factors and outcomes related to pregnancy.

We acknowledge that these findings may initially seem contradictory, but they emphasize the need for a nuanced understanding of the mechanisms involved. Our study provides evidence of both positive and negative impacts of vaginal sex on sETB, highlighting the intricate nature of this relationship.

We have included the explanation of these findings in the revised discussion.

  1. Also, the discussion section of the manuscript is insufficient. Please discuss your findings;explain the significance of those results and tie everything back to the research question. Interpretation of the results is inconsistent. Practical implications and future research direction are not mentioned. Please discuss the generalisability (external validity) of the study results. In conclusion, clarify the message that the reader will take home.Is vaginal sex safe or not? Moreover, lots of articles in the literature investigated the relationship between vaginal sex and preterm birth. What is your contribution to the current literature?

Response: Thank you so much for the suggestions. We have rewritten the discussion section which addressed the concerns you raised. We hope that this revised discussion provides a well-rounded conclusion that captures the key points and implications of our study.

Comments on the Quality of English Language

- There are several typographical errors that should need to be corrected (i.e. factor-alpha, we hypothesize that vaginal sex, this stıdy has the potential, early-term birth, etc.).

Response: Thanks for your careful readings. We have fixed all typos in the revised version.

Reviewer 2 Report

This study which is retropective in nature determines the effect of sexual intercourse on obstetric outocomes.

The title includes only Black women. Some supporting evidence should be stated for having included only this sector of the population. The abstract is comprehensively written.The 'Introduction' is appropriate but should make reference to the role of socio-economic factors that influence preterm births and preterm premature rupture of membranes. 

Method: The study design could include a flow chart to help the reader. Details of recruitment of subjects and documentation of informed consent should be stated. What was the sample size? Good to show how this was calculated. Dating of pregnancy with LMP and US is stated . One vital parameter would be length of cervix. Kindly be clear about inclusion and exclsuion criteria in selection of subjects. Clearly patients who had PPROM , preterm labour and other obstetric complications that would affect onset of labour would have been excluded.

In describing the blood and cervico-vaginal specimens, kindly provide the techniques of collection . Were samples taken from the endocervix, upper vagina and lower vagina-these are essential as vaginal flora varies by sites.

Results. 

The tables and figures are well laid out and supports the discussion. It would be good to also include the birth details - neonatal outcomes. Admission to SCN and neonatal sepsis may be used as surrogates of on-going infection in mothers.

Discussion: 

The discussion needs to be expanded as some facts stated in Introduction are repeated. The complex relationship between sexual intercourse and obstetric outcomes is alluded to . Agreeably vaginal inflammation plays a role. Other factors like short cervical length , nutritonal status, parity and socio-economic status are some factors to consdier. These are limitations of the study.

Ethics approval; Kindly indicate if this study has been approved by the Institutional Review Board.

References: Check with the journal requirements for formatting. 

Author Response

We highly appreciate the reviewers’ constructive and valuable comments which significantly improve our manuscript. We now provide point-to-point response to the comments, with the changes highlighted in RED in the manuscript.

Reviewer 2 Comments and Suggestions for Authors

This study which is retrospective in nature determines the effect of sexual intercourse on obstetric outcomes. The title includes only Black women. Some supporting evidence should be stated for having included only this sector of the population. The abstract is comprehensively written. The 'Introduction' is appropriate but should make reference to the role of socio-economic factors that influence preterm births and preterm premature rupture of membranes.

Response: Thanks for the suggestion. We note that this is a secondary data analysis project. We included the Black women in the title to highlight the fact that study samples from the original study were Black women only. We agree that the socio-economic factors play important roles in preterm birth and preterm premature rupture of membranes. In the revised version, we have included additional references to the role of socio-economic factors in both the introduction and discussion sections.

“Among the known risk factors for preterm birth (PTB) in the United States, one of the most significant is maternal race, particularly Black race.17 According to the most recent US Vital Statistics report on PTB, the overall PTB rate in the country is 10.1%.18 However, when we consider racial breakdown, a more nuanced picture emerges, revealing a PTB rate of 14.4% among Black women, which is over 50% higher than the rate of 9.1% among White women. This disparity is most pronounced for spontaneous preterm birth (sPTB), occurring four times more frequently in Black women (8.9%) compared to White women (2.2%). Consequently, this contributes to the higher incidence of low birth weight among Black infants, who are twice as likely as White infants to have low birth weight (14% vs. 7%) and nearly 2½ times more likely to experience infant mortality.19 While low socioeconomic status (SES) is recognized as a risk factor for PTB, studies consistently demonstrate that SES alone explains less than half of the Black-White disparity in PTB. 1, 20

Method: The study design could include a flow chart to help the reader. Details of recruitment of subjects and documentation of informed consent should be stated. What was the sample size? Good to show how this was calculated. Dating of pregnancy with LMP and US is stated. One vital parameter would be length of cervix. Kindly be clear about inclusion and exclsuion criteria in selection of subjects. Clearly patients who had PPROM, preterm labour and other obstetric complications that would affect onset of labour would have been excluded.

Response: Thank you for asking this question. This is a secondary data analysis project. We are not directly involved in the data collection process. Study inclusion and exclusion criteria have been described elsewhere (Corwin 2017 BMC Pregnancy Childbirth, PMID: 28571577). Regardless, we have now included the selection criteria in the original study in the Methods section (see below).

“Selection Criteria. Participants were included in the original study33 if they 1) self-identified as African American, 2) were between 8-14 weeks gestation (gestational age was confirmed by clinical records and/or ultrasound), 3) able to comprehend written and spoken English, 4) between 18-40 years of age, and 5) were not experiencing any chronic medical conditions or taking prescriptions for chronic conditions (verified by prenatal records). Additionally, participants needed to live within a 20-mile radius of the laboratory, to decrease the amount of time spend transporting the biological samples, thus preserving the validity of the biomarkers.”

In describing the blood and cervico-vaginal specimens, kindly provide the techniques of collection.Were samples taken from the endocervix, upper vagina and lower vagina-these are essential as vaginal flora varies by sites.

Response: Thanks for the question. We have included the sample collection techniques in the revised version (see below).  

“The study coordinator escorted each woman individually to a private room, where they were provided her both verbal and pictorial instructions on self-collecting vaginal swabs. After answering any questions, the research coordinator stepped out of the room while the woman privately inserted the swab approximately 3-4 inches from the introitus, into the midportion of the vaginal vault. Vaginal sampling was done using a sterile Catch-AllTM Sample Collection Swab (Epicentre Biotechnologies, Madison WI). After swabbing, and without touching the top of the swab, the woman immediately handed it to the research coordinator who then placed the swab in a MoBio bead tube (moGio Laboratories, Inc.). This tube was clearly labeled and kept on ice until transferred to the lab where it was immediately stored at -80o until assayed, including for cytokines using standard enzyme-linked immunoassay (ELISA) kits (R&D Systems, cat# SCRP00). The concentration of C-reactive protein (CRP) was determined via the four parameter calibration curves identified using the BioTek Gen5 software.”

Results. The tables and figures are well laid out and supports the discussion. It would be good to also include the birth details - neonatal outcomes. Admission to SCN and neonatal sepsis may be used as surrogates of on-going infection in mothers.

Response: Thank you for the suggestion. Since this is a secondary data analysis project, the additional information on neonatal outcome, admission to SCN, and neonatal sepsis were not included in the data we requested. We are not sure whether information on neonatal outcome, admission to SCN, and neonatal sepsis have been collected in the original study. Regardless, we appreciate the reviewer’s thoughtful suggestion.   

Discussion: The discussion needs to be expanded as some facts stated in Introduction are repeated.The complex relationship between sexual intercourse and obstetric outcomes is alluded to . Agreeably vaginal inflammation plays a role. Other factors like short cervical length, nutritonal status, parity and socio-economic status are some factors to consdier. These are limitations of the study.

Response: Thank you for the suggestions. We have expanded the discussion section. In response to this specific comment, we have added the following limitation to the discussion section.

“Another limitation of this study is that we did not explore the role that cervical length, and nutritional status play on birth outcomes. Future research should examine the relationships between these factors and gestational age at birth outcomes and their relationships with vaginal sex.”

Ethics approval; Kindly indicate if this study has been approved by the Institutional Review Board.

Response: Our study is a secondary data analysis project, and by NIH definition, this is not a human subject study. Regardless, we added the following to the method section.

Ethics Approval. This is a secondary data analysis study, and the institutional review board ethics approval was obtained for the original study34.

References: Check with the journal requirements for formatting.

Response: Thank you for pointing it out. We have changed the ACS reference style in the revised version.  

Reviewer 3 Report

First of all, I would like to congratulate and thank you for the work behind this paper. I would like to make several comments about your manuscript:

  1. In the statistical analysis section, you do not clearly state which normality tests you have used to verify the use of parametric tests for hypothesis testing. Please perform a normality test and address this point. Additionally, when you do so, please add it to the manuscript in this section. If normality is not present, use non-parametric tests for hypothesis testing.
  2. On what date were the data collected?
  3. Is the sample used representative of the African American population in the United States? Why did you focus on collecting data from the African American population?
  4. What criteria did you use to determine the cutoff point for late pregnancy in the range of 24-30 weeks of gestation? Do you consider a gestation between week 24 and 30 to be considered late?
  5. How was the sampling conducted? What sampling techniques were used to collect the data?
  6. Please include the logistic regression table in the manuscript.
  7. Table 1 should include explanations of the acronyms and abbreviations used. Some terms used are not clear.
  8. Please rewrite the discussion section. There are paragraphs that are too lengthy and make it difficult for the reader to understand the conclusions. Additionally, be cautious when generalizing the conclusions as the variable of interest, vaginal sexual intercourse, is not properly quantified and is analyzed only as a dichotomous variable.
  9. In the limitations section of the study, make it clear the challenging quantification of the "sexual intercourse" variable. This is actually a significant limitation as the results may differ between women who engage in occasional vaginal intercourse and those who have regular vaginal intercourse. How were these data lost? What difficulties were encountered in collecting these data? Do you plan for further research taking into account the frequency of sexual intercourse? What new lines of research do you propose after this study?

Author Response

We highly appreciate the reviewers’ constructive and valuable comments which significantly improve our manuscript. We now provide point-to-point response to the comments, with the changes highlighted in RED in the manuscript.

Reviewer 3 Comments and Suggestions for Authors

First of all, I would like to congratulate and thank you for the work behind this paper. I would like to make several comments about your manuscript:

Thank you for the positive comments, and we appreciate your constructive and insightful comments.

In the statistical analysis section, you do not clearly state which normality tests you have used to verify the use of parametric tests for hypothesis testing. Please perform a normality test and address this point. Additionally, when you do so, please add it to the manuscript in this section. If normality is not present, use non-parametric tests for hypothesis testing.

Response: When the cytokine measures are treated as outcomes, linear regression models were used. In this case, we checked the normality assumption of the models by several approaches including by plotting a histogram of the residuals, by QQ plots, and by Shapiro-Wilk test. No violation of the normality assumption was noted. We added the following to the statistical analysis section.

“Residual plots and Shapiro-Wilk test were used to check potential violation of models assumptions.”

On what date were the data collected?

Response: Thanks for the question. The data was collected between 2014 and 2018. We have clarified the data collection dates in the Methods section (copied below).

“Overview. This study is a secondary analysis of a 5-year prospective longitudinal study conducted in Atlanta, Georgia. The original study was conducted between 2014 and 2018 and enrolled over 500 healthy, Black women in their first trimester of pregnancy and followed them through delivery, as described in detail elsewhere.33 Data were collected prospectively during routine prenatal care visits at two time points: the first visit occurred between 8-14 weeks gestation, and the second visit between 24-30 weeks’ gestation. This study design allowed for the assessment of various factors, including vaginal sex and inflammatory markers, in real-time as they occurred during the participants' pregnancies. Medical records were used to determine infant gestational age at birth using the American College of Obstetricians and Gynecologists (ACOG??) algorithm, which combines ultrasound and last menstrual period information.34” 

Is the sample used representative of the African American population in the United States?

How was the sampling conducted? What sampling techniques were used to collect the data?

Response: The sample used is not representative of the African American population in the United States. As described in the original study34, the cohort included pregnant US-born women who self-identified as African American (AA) and who were receiving care from one of two hospitals in Atlanta, GA: a private hospital affiliated with Emory University and a county-supported hospital also in Atlanta, GA that primarily provided prenatal care to the low-income and underserved women. Women eligible for recruitment were those between the ages of 18-40 presenting for a prenatal visit between weeks 8-14 of gestation, anticipating a singleton pregnancy later verified by review of the clinical record, and who met inclusion criteria of being 1) US-born AA by self-report; 2) generally healthy with no chronic medical condition and not regularly taking any medications with the exception of prenatal vitamins; 5) able to comprehend written and spoken English; 6) interested in hearing about a prenatal study. In this sense, the sample used is not representative of the African American population in the United States. We have now included this as a limitation of the study.

Why did you focus on collecting data from the African American population?

Response: We note that this is a secondary data analysis project. We were not directly involved in the recruitment process. Based on what we learned, the original study design was based on a health disparity research framework. This study design allows researchers to better explain health disparity by first looking within the high burden group to identify intra-group risk and protective factors (Rowley et al., 1993, PMID: 8123282).

What criteria did you use to determine the cutoff point for late pregnancy in the range of 24-30 weeks of gestation? Do you consider a gestation between week 24 and 30 to be considered late?

Response: Thank you for this important question. When we designed the study, we wanted to collect survey and biosamples late enough to capture most of the second trimester while still being early enough to capture data before onset of preterm births. Therefore, we decided to use 24-30 weeks as cutoff point for late pregnancy.

Please include the logistic regression table in the manuscript.

Response: The logistic regression Table with sETB as the outcome is now included in the revised version (see Table 2).

Table 1 should include explanations of the acronyms and abbreviations used. Some terms used are not clear.

Response: Thanks for bringing this to our attention. We have included explanations of the acronyms and abbreviations in the revised version and made sure all termed were clear.

Please rewrite the discussion section. There are paragraphs that are too lengthy and make it difficult for the reader to understand the conclusions. Additionally, be cautious when generalizing the conclusions as the variable of interest, vaginal sexual intercourse, is not properly quantified and is analyzed only as a dichotomous variable.

Response: Thank you for the suggestions. We have rewritten the discussion section to address the concerns you raised.

In the limitations section of the study, make it clear the challenging quantification of the "sexual intercourse" variable. This is actually a significant limitation as the results may differ between women who engage in occasional vaginal intercourse and those who have regular vaginal intercourse. How were these data lost? What difficulties were encountered in collecting these data? Do you plan for further research taking into account the frequency of sexual intercourse? What new lines of research do you propose after this study?

Response: Thank you for the comment. We have added the followings to the limitation section.

“Finally, one major limitation of our study is that the main exposure variable, vaginal sexual behaviors, was coded as a binary variable. It has been reported that vaginal cytokine levels can be affected by various sexual practices such as the number of partners, frequency of vaginal sex, and unprotected vaginal intercourse.46-48 Although we collected data on the frequency of vaginal sexual behaviors at each trimester, there were missing data on these measures due to participants ‘skipping’ or not providing an answer to this item, precluding our ability to study the dose-response relationship between vaginal sexual behavior and the risk of sETB. Future research should explore the role of vaginal sex on ETB with vaginal sex as a frequency as opposed to a binary variable. Exploring vaginal sex through the lens of frequency will provide a more granular understanding of the influence of vaginal sex in terms of whether there are any potential differences in frequent or occasional vaginal sex during the later weeks of pregnancy on gestational age at birth outcomes.”

Round 2

Reviewer 1 Report

The findings are contradictory.

Author Response

Response: We thank the reviewer for raising this concern. We respectfully disagree with the reviewer 1’s comment and explained further in the manuscript why our findings are not contradictory. We hope the revised text will help clarify the confusion that the reviewer 1 had. Changes have been made in several places and are marked in red. FYI, we copied below the main changes.

In the Results Section:

“Figure 2 shows the mediation analysis results, and regression coefficients from these models can be found in Table 2. First, there was a significant overall positive effect of vaginal sex during late pregnancy on sETB (β = -0.944, OR = 0.39, p-value = 0.003). Controlling for potential confounding effects due to age, sex of newborn, parity, marital and cohabitate status, and education level, our model indicated that, the odds of having sETB for women with vaginal sex at late pregnancy were 61% lower than that for those without vaginal sex. Second, in the mediation pathway, we observed that having vaginal sex was associated with a 0.63 unit increase in log ratio of IL-6/IL-10 (β=0.63, p-value = 0.036). Furthermore, a one-unit increase in the log ratio of IL-6/IL-10 was associated with an increase in the odds of having sETB by 19.9% (β = 0.1812, p = 0.02). Mediation analysis suggested that engaging in vaginal sex could lead to an increase in IL-6 levels, which, in turn, could contribute to an increased likelihood of experiencing sETB.

In summary, although engaging in vaginal sex was associated with an overall significant reduction in the odds of experiencing sETB (total effect = - 0.1560, p-value = 0.015), it is important to recognize the potential offsetting effects (indirect effect = 0.0313, p-value = 0.026, proportion mediated = 18.90%). This implies that approximately 18.9% of the total effect of vaginal sex on sETB can be attributed to the mediating role of IL-6, indicating a partial mediation effect in the relationship between vaginal sex and sETB. It is understandable that IL-6, as an inflammatory biomarker, may not completely explain the relationship between vaginal sex and sETB, given that sETB has various causes. Therefore, it is crucial to recognize the multifactorial nature of sETB and consider other contributing factors in future investigations.”

In the Discussion Section:

“Second, this analysis builds upon the literature surrounding the influences on sETB, which is a phenomenon that impacts the health of newborn but is not studied as often as sPTB. The full impact of vaginal sex on ETB requires further research. Towards this goal, our study tested two hypotheses: 1) whether vaginal sexual behavior during pregnancy increase the odds of having sETB, and 2) whether the negative impact of vaginal sexual behavior on sETB is mediated by the inflammatory biomarkers. Our analysis showed an overall significant positive effect of vaginal sex during late pregnancy on sETB, indicating a reduced risk of having sETB. Interestingly, mediation analysis suggested that vaginal sexual behavior could have a negative impact on sETB, as vaginal sex during late pregnancy was associated with an increased vaginal IL-6/IL-10 ratio, while increased vaginal IL-6/IL-10 ratio was linked to an increased risk of sETB.

The mediation analysis result was in line with our original hypothesis that vaginal sex has a negative impact on the sETB. Previous research has established the importance of IL-6 in triggering spontaneous preterm birth, indicating that elevation of IL-6 level may be involved in regulating the onset of labor55 through its positive correlation with the expression of genes controlling prostaglandin synthesis and signaling.56 Recent studies have also reported elevated cervico-vaginal fluid concentrations of IL-6 among women with preterm delivery.57 Our study adds to this body of evidence by linking vaginal sex during late pregnancy with an increase in vaginal IL-6/IL-10 ratio, which in turn results in an increased risk of having sETB. Taken together, these findings suggest that vaginal IL-6. These findings shed lights on the significance of IL-6 as a potential biological pathway through which vaginal sex influences the likelihood of experiencing sETB.

It is important to note that the overall significant positive (i.e., beneficial) effect of vaginal sex on sETB was contradictory to what we have initially hypothesized. However, this seemingly contradictory result was not surprising and explainable. In this study, we have an inconsistent mediation model58 (i.e., positive indirect effect and negative direct effect, see Figure 2). In the field of medical research, it is not uncommon to encounter inconsistent mediator effects, which can be particularly significant when assessing counterproductive effects of experiments. In such cases, the manipulation involved may have resulted in contradictory mediated effects. For a more comprehensive understanding of this topic, interested readers are encouraged to refer to the work of Mackinnon et al.58 and the references provided therein for valuable insights into the complexities associated with inconsistent mediator effects in medical research. Briefly, relevant to our study, inconsistent mediator effects can occur when there are different positive and negative pathways through which vaginal sexual activity influences sETB. Indeed, the relationship between sexual intercourse and obstetric outcomes is multifaceted, likely influenced by various factors beyond vaginal inflammation. While our study focused only on the role of vaginal inflammation, it is crucial to acknowledge the potential contribution of other factors in the complex relationship between vaginal sexual behavior and birth outcomes. For example, previous research has suggested that factors such as short cervical length 59, 60, poor nutritional status61, higher parity62, 63, and lower socioeconomic status 64 could impact birth outcomes. Health benefits of vaginal sexual activity on a wide range of medical and psychological measurements have also been reported65. As demonstrated by our inconsistent mediation model, this investigation suggested that there might be multiple potential mechanisms by which vaginal sexual activity positively or negatively influences the birth outcome.  Therefore, to provide a more comprehensive understanding of the relationship between sexual intercourse and obstetric outcomes, future studies should take into considerations of a broader set of risk or protector confounders/mediators, enabling a more nuanced interpretation of the association between vaginal sex and birth outcomes.”

Reviewer 3 Report

God job

Author Response

Thank you so much for the positive review! We are grateful for your insightful suggestions in the previous round.